# Tackling Complexity: Integrating Responses to Internal Displacements, Extreme Climate Events, and Pandemics

Roberto Ariel Abeldaño Zuñiga [1,2,*], Gabriela Narcizo de Lima [3,4] and José Carlos Suarez-Herrera [5]

1 Yhteiskuntadatatieteen Keskus, Valtiotieteellinen Tiedekunta, Helsingin Yliopisto, 00150 Helsinki, Finland
2 Helsinki Institute of Urban and Regional Studies (URBARIA), University of Helsinki, 00150 Helsinki, Finland
3 Geography Department, Faculty of Arts and Humanities, Porto University, 4150-564 Porto, Portugal; gabrielalima@letras.up.pt
4 Centre of Studies in Geography and Spatial Planning, Porto University, 4150-564 Porto, Portugal
5 Office of Research and Knowledge Transfer, Mid-Atlantic University, 35017 Las Palmas de Gran Canaria, Spain; josecarlos.suarez@pdi.atlanticomedio.es
* Correspondence: ariel.abeldanozuniga@helsinki.fi

**Abstract:** Background: During 2020 and 2021, over 50.2 million individuals were forced to leave their homes to escape the impacts of climate-related disasters, unable to practice social isolation or self-quarantine. A considerable proportion of them reside in densely populated areas with a lack of basic services such as water and sanitation and limited access to essential healthcare. This study aimed to estimate the internal displacements during 2020 and 2021 due to climate-related events, and review the evidence for proposing policy recommendations. Methods: Data from the Internal Displacement Monitoring Centre were used for assessing internal displacement by disasters during 2020 and 2021. In addition, the authors conducted a bibliographic review to analyse the responses to internal displacements in climate-related disasters. Results: There were 883 severe storms and 1567 flood events resulting in 50.2 million internal displacements globally. Through the documents reviewed, the legal framework, the vulnerabilities and current challenges of internally displaced persons, and the response policy recommendations were analysed. Conclusions: The increased awareness of displacement and migration, particularly driven by climate-related factors, aligns with international agreements emphasising coordinated action. This recognition becomes even more critical in the context of the convergence of climate-related displacements and the potential for future pandemics.

**Keywords:** internal displacements; climate-related disasters; COVID-19; pandemics; disaster risk reduction

## 1. Introduction

Published reports have detailed the increased risk faced by internally displaced people during the global coronavirus (COVID-19) pandemic [1,2]. Over 50.2 million individuals were forced to leave their homes to escape the impacts of climate-related disasters, unable to practice social isolation or self-quarantine. A significant proportion of them reside in densely populated areas with a lack of basic services such as water and sanitation and limited access to essential healthcare.

The extreme climate events that led to forced displacement directly impacted regions already affected by the pandemic, exacerbating health risks and challenging response capabilities. The massive mobility of people in disaster situations heightened the vulnerability of these individuals and overwhelmed healthcare systems. This connection underscores the need for integrated approaches and global resilience to address the complex interactions between climate-induced displacement and the management of health crises such as the COVID-19 pandemic. Although the COVID-19 pandemic may have ended in 2023, the relevance of the issue persists due to its lasting impacts and the ongoing need to address the

intricate interconnections between extreme climate events and health crises. The climate emergency continues to trigger massive displacements, affecting vulnerable communities worldwide. The lessons learned during the pandemic emphasise the importance of a global and coordinated response to crises. Moreover, the post-pandemic world requires resilient strategies to simultaneously address multiple challenges, such as climate-induced displacements, emphasising the need to sustain attention and action on this relevant issue.

The humanitarian impact of internal displacements caused by extreme climate events is significant and manifests in many ways [3]. Affected communities not only experience the loss of homes and belongings but also the disruption of social networks and support systems. This situation heightened the vulnerability of displaced individuals as access to basic services, such as clean water and healthcare, was severely disrupted or limited. The precarious living conditions in temporary shelters increased susceptibility to other diseases, not only to COVID-19 but also to the continuity of care for other non-communicable chronic diseases. Moreover, communities already facing economic and social challenges were further adversely affected, creating a cycle of humanitarian challenges. Highlighting these humanitarian impacts in past years is crucial to raise awareness about the need for preparedness to provide timely humanitarian responses to future challenges. This study aimed to (1) estimate the internal displacements during 2020 and 2021 due to climate-related events, and (2) review the evidence for proposing policy recommendations. So, this study underscores how the convergence of these events directly affected individuals, their families, and communities. It emphasises the importance of addressing not only immediate needs but also working towards long-term solutions that strengthen community resilience and mitigate future risks.

## 2. Materials and Methods

Data from the Internal Displacement Monitoring Centre (IDMC) based in Geneva, Switzerland, were used for assessing internal displacement by disasters during 2020 and 2021. Together with the Norwegian Refugee Council, this body monitors the situation of internal displacement due to circumstances of violence and disaster [4]. For this study, the authors retrieved all internal displacements caused by disaster situations recorded in the IDMC database.

The IDMC is the leading NGO for collecting, processing, curating, and publishing information on internal displacement due to disasters and conflicts globally. Their methods and terminology have been clearly described on their website [4]. The IDMC prioritises its data sources, with the first level comprising data primarily produced by government agencies, UN organisations, the International Organization for Migration (IOM), humanitarian clusters, and local authorities. The second level includes data from international and local NGOs, civil society, human rights organisations, and academia. Finally, the third level contains data from international and local media, affected populations, and non-state armed groups [4]. The first-level sources are considered the most reliable and are used as the primary basis for producing estimates. The second-level sources are consulted only when information from first-level sources is unavailable. The third-level sources are only used if their reliability can be rigorously assessed and verified [4].

According to the definition of internal displacement by the IDMC, this term refers to the "estimated number of internal displacement movements to have taken place during the year. Figures include individuals who have been displaced more than once. In this sense, the number of internal displacements does not equal the number of people displaced during the year" [4].

In addition to the database consulted for conducting the analysis of internal displacement by disasters mentioned above, an extensive bibliographic review was also carried out. Due to the specificity of the topic of responses to internal displacement in climate-related disasters, the selection of documents for the corpus was based on intentional criteria. The papers, book chapters, reports, and institutional reports were traced using various sources. Websites of governmental agencies and international organisations (WHO/OMS,

PAHO/OPS, UNISDR/EIRD) were searched to gather data on documents related to the search terms "internal displacements" and "disaster situations". The period selected was 2020–2021, and the languages of the documents searched are Spanish, English, Portuguese, and French.

The resulting documents were subjected to a screening process to review the relevance of each article (and associated keywords) to the object of study. Documents were included if they represented original research, government documents, or literature reviews on the topics cited and explicitly documented these concepts in the body of the manuscript.

To assess the articles that met the inclusion requirements, the research objectives and results were summarized. The argumentation of this article was structured in light of the results provided by the selected articles and related to our categories of analysis. To organise the report of the results, the authors created categories related to the responses: (1) internal displacements during 2020 and 2021 due to climate-related events, (2) legal framework, (3) vulnerabilities and current challenges of internally displaced persons, and (4) response policy recommendations.

## 3. Results

### 3.1. Internal Displacement during 2020 and 2021 Due to Climate-Related Events

The analysis of data from the Internal Displacement Monitoring Centre [4] and the bibliographic review addressing responses to internal displacement in climate-related disasters has yielded relevant results. However, it is important, initially, to define some of the concepts discussed for a comprehensive understanding of the results.

Climate change refers to long-term alterations in global climate patterns, primarily resulting from human activities releasing greenhouse gases into the atmosphere. These changes have significant impacts on climate, such as rising average temperatures, alterations in rainfall patterns, and more frequent extreme weather events, directly influencing ecosystems, biodiversity, and the daily lives of communities. Extreme climate events, in turn, are intensified and are uncommon episodes within the climate system. They encompass phenomena like severe storms, heatwaves, floods, wildfires, and hurricanes, causing significant impacts on the environment, infrastructure, and society. These events often lead to disasters, population displacements, loss of human lives, and substantial damage to ecosystems. Understanding these concepts is crucial for addressing the complex and interconnected challenges related to the environment and climate, seeking adaptation and mitigation strategies to ensure the resilience of communities in the face of these adverse scenarios [5,6].

According to data recorded by the Internal Displacement Monitoring Centre [4], during the years 2020 and 2021, there were 883 severe storm events worldwide (including cyclones, hurricanes, and typhoons) and 1567 flood events resulting in forced displacements of people. These events impacted 160 countries, with a more significant impact in regions of the Global South, such as Central America, Sub-Saharan Africa, and East and Southeast Asia. In total, these events led to 50.2 million internal displacements globally (Figure 1), against the backdrop of the health crisis stemming from the COVID-19 pandemic. Table 1 displays the top twenty events with the highest records of internal displacements during 2020 and 2021 [4]. It can be observed that these events cumulatively accounted for over 28.3 million displacements, impacting countries in Asia, Central America, and Africa most significantly. The interplay of climate-related disasters and the recurrent waves of the pandemic during those years has exacerbated the challenges faced by displaced populations and required comprehensive policy responses.

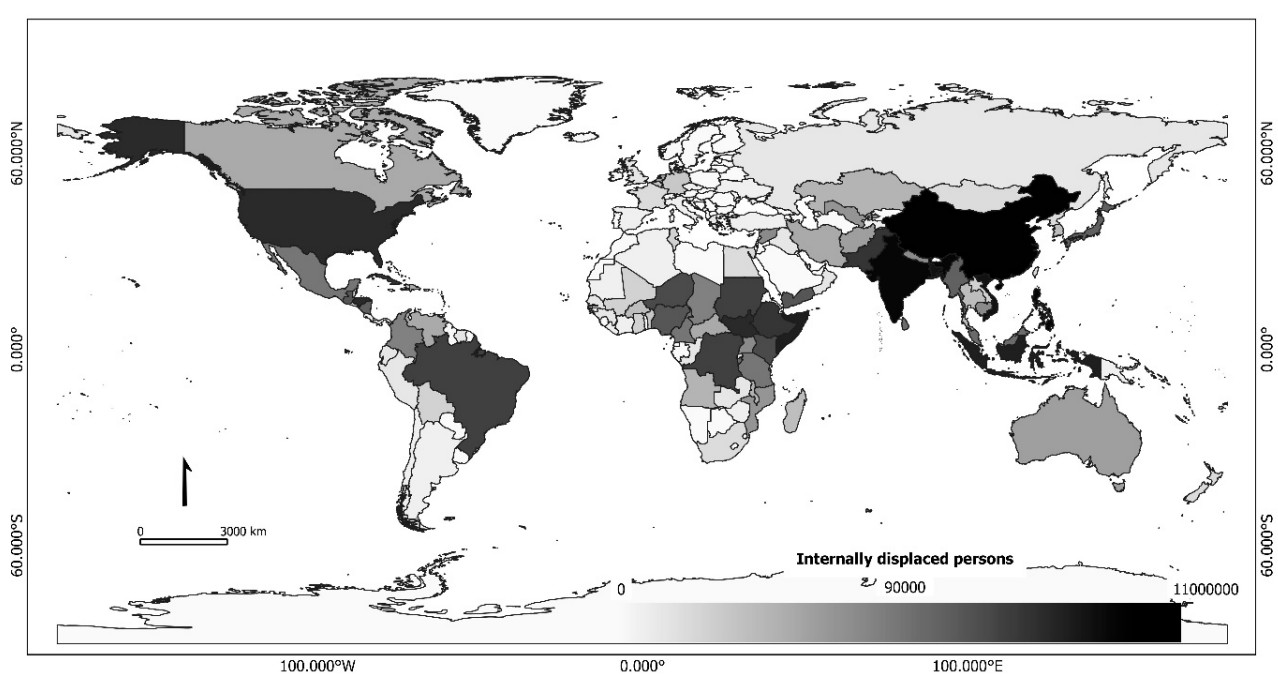

**Figure 1.** Internal displacements resulting from climatic events—2020 to 2021.

**Table 1.** Top twenty climate-related events with the highest records of internal displacements during 2020 and 2021.

| N | Affected Countries and Territories and Event Name | Date of Event (Start) | Internal Displacements |
|---|---|---|---|
| 1 | India, Bangladesh, Myanmar, Bhutan: Cyclone Amphan | 16 May 2020 | 4,950,000 |
| 2 | The Philippines, Palau, Viet Nam: Typhoon Rai (known as Odette in the Philippines) | 11 Dec 2021 | 3,915,000 |
| 3 | China: Flood Season | 1 Jun 2020 | 3,760,000 |
| 4 | Bangladesh: Flood | 26 Jun 2020 | 1,921,000 |
| 5 | The Philippines, Viet Nam: Typhoon Vamco (known as Ulysses in the Philippines) | 8 Nov 2020 | 1,560,000 |
| 6 | India, Bangladesh: Tropical Cyclone Yaas | 23 May 2021 | 1,505,000 |
| 7 | China: Flood | 16 Jul 2021 | 1,490,000 |
| 8 | China, The Philippines, Taiwan: Typhoon "In-Fa"/"Fabian" | 14 Jul 2021 | 1,430,000 |
| 9 | The Philippines, Vietnam: Typhoon Goni (known as Rolly in the Philippines) | 26 Oct 2020 | 1,250,000 |
| 10 | Pakistan: Flood | 15 Jun 2020 | 810,000 |
| 11 | Colombia, Nicaragua, Honduras, El Salvador, Guatemala, Belize: Hurricane Iota | 14 Nov 2020 | 743,000 |
| 12 | The Philippines, Viet Nam: Tropical Cyclone Conson (known as Jolina in the Philippines) | 6 Sep 2021 | 727,000 |
| 13 | India, Bangladesh: Tropical Cyclone Yaas | 23 May 2021 | 703,000 |
| 14 | The Philippines, Viet Nam: Typhoon Molave (known as Quinta in the Philippines) | 23 Oct 2020 | 623,000 |
| 15 | Dominican Republic, Haiti, Cuba, Puerto Rico, USA: Hurricane Laura | 21 Aug 2020 | 585,000 |
| 16 | Somalia: Gu rains | 31 Mar 2020 | 505,000 |
| 17 | India: Flood | 1 Jun 2021 | 500,000 |
| 18 | China: Floods | 10 Aug 2020 | 490,000 |
| 19 | China: Flood | 1 Sep 2021 | 467,000 |
| 20 | Sudan: Flood | 1 Jul 2020 | 454,000 |
| | **Total Internal displacements** | | **28,388,000** |

*3.2. Legal Framework*

In the existing legal framework for the protection of internally displaced persons, international instruments such as the Guiding Principles on Internal Displacement can be found, developed by the UN Secretary General's Representative on Internal Displacement [7]. These principles underscore the responsibility of states to provide protection and assistance to internally displaced persons within their borders.

The awareness of displacement and migration resulting from disasters, climate change, and environmental degradation has grown significantly since 2010, leading to discussions at various levels. This recognition is reflected in international legally binding agreements that emphasise the necessity for coordinated, cross-sectoral action [8].

One crucial agreement is within the United Nations Framework Convention on Climate Change. Migration issues were initially incorporated into the UNFCCC in the 2010 Cancun Adaptation Framework, gaining further anchoring with the 2015 Paris Agreement [9]. The Paris Agreement set up a task force on Displacement under the Executive Committee of the Warsaw International Mechanism for Loss and Damage Associated with Climate Change [8].

In addition to legally binding agreements, several international policy frameworks integrate references to displacement and migration prompted by disasters, climate change, and environmental degradation. The Sendai Framework for Disaster Risk Reduction 2015–2030 [10] seeks to minimise the impact of disasters on people, the environment, and economies, emphasising action in economic, social, and environmental policy areas.

The Global Compact for Safe Orderly and Regular Migration of 2018 [11] represents a critical advancement in global governance on environmental migration. It includes specific actions to address the adverse drivers and structural factors compelling people to move due to natural disasters, climate change, and environmental degradation.

The 2030 Agenda for Sustainable Development recognises the serious threat of climate change and environmental degradation to the achievement of Sustainable Development Goals [12]. Leveraging investments in activities addressing both sustainable development and climate action is highlighted as an opportunity.

Initiatives like the Nansen Initiative's Agenda for the Protection of Cross-Border Displaced Persons [13] and the Platform on Disaster Displacement [14] provide a coherent approach to the protection of people displaced across borders in the context of disasters and climate change.

Recent reports, including the United Nations Secretary General's High-Level Panel on Internal Displacement [15] and the Action Agenda on Internal Displacement [16], emphasise proactive and systematic approaches to addressing internal displacement as part of the UN's work on climate change.

Regionally, specific agreements are observed, such as the African Union's Kampala Convention on Internal Displacement in Africa [17] and the American Convention on Human Rights in the Americas [18]. These treaties aim to ensure fundamental rights for internally displaced persons, including protection against inhumane living conditions, but they are not the only ones, as many countries have developed national policies and legislation to address issues related to internal displacement.

Collectively, these agreements and frameworks underscore the urgency of addressing displacement caused by climate change and disasters. They emphasise the need for coordinated efforts, legal preparedness, and global cooperation to protect affected populations and build resilience in the face of environmental challenges. However, the applicability of these legal frameworks in the context of pandemics or epidemics is a challenge. The response to internal displacement must be adjusted to address the complexity of multi-threat scenarios. It is essential to consider how public health measures impact the specific rights and needs of internally displaced persons, such as access to safe shelter, continuous medical care, and basic services. The assessment of current protection should analyse the flexibility of existing legal frameworks to adapt to health crises, ensuring that internally displaced persons are not only protected from disease but also from the additional vulnerability

associated with displacement. Collaboration between government entities, humanitarian organisations, and civil society is crucial to ensure a comprehensive and legally robust response in these complex scenarios.

Legal preparedness plays a crucial yet often overlooked role in effective disaster risk management. In the aftermath of a disaster, time becomes a critical factor, and having well-established laws in place is essential. Laws can be instrumental in addressing the complex challenges arising from climate-related displacement, particularly when contemplating planned relocation as a proactive measure to mitigate the risk of further displacement due to climate-related events or as a durable solution for existing displacement. Nevertheless, some legal gaps have been identified that should be addressed to strengthen the protection of internally displaced persons in the context of pandemics and epidemics, such as the lack of specific provisions considering the unique needs of this population during health crises.

*Adaptation of Guiding Principles to Health Crises*: The Guiding Principles on Internal Displacement may not adequately address circumstances arising from a pandemic. Therefore, it is necessary to review and adapt these principles to include specific provisions ensuring the protection and healthcare of internally displaced persons in public health emergencies.

*Specific Regional Legal Framework*: Some regions may lack a specific legal framework addressing the complexities of protecting internally displaced persons in the context of pandemics. Therefore, it is necessary to develop or strengthen regional treaties that consider the intersections between displacement and health crises and adapt existing regulations.

Intensifying actions to prevent and minimise: Governments need to broaden their efforts to prevent, minimise, and address climate-related displacement. However, this necessitates a review and strengthening of existing laws to ensure that such actions are grounded in solid legal foundations.

*Promoting early action*: The promotion of actions before displacement, including climate change adaptation and early measures, requires governments to incorporate clear and specific legal provisions to guide such initiatives.

*Protection against Discrimination*: Current regulatory frameworks may not offer sufficient protection against health-related discrimination of internally displaced persons. A potential solution would be to incorporate explicit provisions ensuring equal access to medical care and other services.

*Addressing protection needs*: Legal frameworks should explicitly address the protection and assistance needs of people displaced due to the adverse effects of climate change. The quest for durable solutions must be legally strengthened, especially for populations experiencing protracted displacement.

*Intersectoral Coordination Mechanisms*: A coordinated approach between sectors, such as health and disaster management, may be lacking in the legal framework. To address this gap, provisions need to be established that encourage collaboration and coordination between various entities to ensure holistic and effective responses [19].

*Increasing Finance for Local Adaptation*: Increasing climate adaptation finance for local organisations requires legal frameworks that promote local engagement and leadership, ensuring that laws support and facilitate the effective use of funds by local entities [20].

*Integration of Climate Finance*: Legal coherence is crucial to ensure the integration of climate finance with humanitarian, disaster risk reduction, and development finance. Donors should operate within established legal frameworks to optimise synergies among these financial streams.

*The International Health Regulations* [21] play a crucial role in managing global health emergencies. These regulations are an international legal instrument aimed at preventing the international spread of diseases and contributing to their control. Although the IHR is not specifically designed to address internal displacement, it can have relevant implications and connections in the context under consideration:

*Global Coordination*: The IHR establishes a framework for the notification and response to events of international public health concern. This promotes global collaboration and

coordination of efforts to address pandemics and epidemics, which is essential when dealing with population displacement associated with health crises.

*Integration with Humanitarian Responses*: In situations where internal displacement is affected by extreme climate events and pandemics, the IHR can provide a framework for integrating humanitarian responses with public health measures. This ensures coordinated and efficient action across different sectors.

*Protection of Public Health*: The regulations aim to protect public health, including the care and protection of people affected by internal displacement during health emergencies. They can influence the formulation of policies and protocols to ensure adequate medical care for this population.

*Adaptation to Local and Mobile Contexts*: These regulations recognise the need to adapt to specific contexts and the mobility of populations. This flexibility may be relevant when considering protocols for internally displaced persons in emergencies. We promote a balance between, on the one hand, a legal framework based on the principles of participatory inclusive governance, and on the other hand, strategic alliances that take shape in the territories between the social actors involved in the structuring of a coordinated global agenda.

In summary, the International Health Regulations provide an international framework that can influence how countries manage situations of internal displacement during health crises. Their effective implementation can contribute to a more coordinated and efficient response at the intersection of displacement, extreme climate events, and pandemics.

### 3.3. Identified Vulnerabilities and Current Challenges

Climate change amplifies existing challenges and vulnerabilities, leading to compounding crises for communities worldwide. Notable instances include Angolan citizens crossing into Namibia in December 2020 due to drought-induced shortages of food, water, healthcare, and employment. In November 2020, Honduras faced consecutive hurricanes (Eta and Iota), causing massive displacement, damaging crops, and worsening the plight of families already suffering from COVID-19 and poverty [22]. Iraq struggles with water shortages, drought, geopolitical tensions, and water mismanagement, placing local communities at risk of displacement and insecurity. In Yemen, a prolonged armed conflict intensified in 2020, coupled with extreme flooding that devastated communities and exacerbated health crises. Over 300,000 people, primarily internally displaced persons, were affected, leading to secondary displacement [20]. These interconnected challenges underscore the urgent need for comprehensive and sustainable climate and humanitarian interventions.

In Central America, specific vulnerabilities are evident in the high exposure to extreme climate events, such as hurricanes and storms, impacting communities in the Caribbean region already prone to social and economic instability [23]. During the pandemic, mobility restrictions limited access to shelters and essential services, exacerbating the vulnerability of these communities. In sub-Saharan Africa, rural populations dependent on agriculture face climate vulnerabilities, while the lack of infrastructure and basic services increases their exposure. The pandemic amplified disparities in healthcare access and challenged response capacities [24]. In Southeast Asia, urban vulnerabilities are accentuated by climate events and high population density. During the pandemic, urban communities experienced disruptions in essential services and faced economic hardships [25]. In all regions, the interaction between contextual vulnerabilities and the pandemic underscored the need to address structural issues to build resilience and prepare for future displacement.

The gaps in responding to internal displacement caused by extreme climate events during the pandemic were conspicuous in various areas. Firstly, mobility restrictions imposed to contain the spread of COVID-19 affected the timely delivery of humanitarian assistance, leaving vulnerable communities without access to shelters, essential supplies, and adequate healthcare. Additionally, the lack of coordination between humanitarian actors and government authorities contributed to inefficiencies in resource distribution. Economic difficulties exacerbated gaps as many displaced individuals lost their jobs and

faced challenges covering basic needs. As we move into the future, gaps persist in terms of preparedness and response capacity, with additional challenges related to forecasting extreme climate events and global coordination. Addressing these gaps is crucial to enhance the resilience of affected communities and ensure more effective responses in similar situations in the future.

### 3.4. Response Policy Recommendations

*Integrated Response Centres*: Establish integrated response centres that operate jointly to address both displacement and health crises. These centres should bring together experts in health, disaster management, coordination, and social services to facilitate a coordinated and efficient response. In 2021, a comprehensive needs analysis was conducted in response to the Rohingya refugee situation in Bangladesh. The purpose was to provide an analysis of how needs have changed in 2020, with an emphasis on the impact of the COVID-19 pandemic on multisectoral needs. The report highlights the need to increase funding for the integrated humanitarian response [26,27].

*Integrated Contingency Plans*: Develop contingency plans that incorporate specific protocols to address situations where displacement and pandemics coexist. These plans should outline specific actions, roles, and responsibilities of the various involved stakeholders. In cases of emergencies, external humanitarian assistance may take days, and in certain instances, weeks, to become available. Hence, it is crucial to establish a preparedness plan grounded in the existing capacity of the country to effectively address the initial stages of an emergency, particularly in regions prone to seasonal weather events like Central America or Southeast Asia. In 2015, the Inter-Agency Standing Committee formulated guidelines for crafting Advanced Preparedness Actions and Contingency Planning, considering the interplay of intricate events involving displaced populations in epidemic or pandemic scenarios [28]. Typically, the fundamental components of a contingency plan—coordinative measures, the composition of aid packages, procurement, and logistics arrangements—are applicable across various potential emergencies within a country. Specialised contingency plans should only be devised when a distinctly different humanitarian response is warranted, for instance, due to significant geographical differences in the affected areas.

*Integrated Surveillance Systems*: Implement integrated surveillance systems that monitor both the spread of diseases and population displacement caused by extreme climatic events. These systems would collect relevant data on disease incidence and population movement patterns. The gathered information would provide a robust foundation for more informed and adaptive decision-making in real time. Furthermore, it would facilitate a more agile and coordinated response, enabling authorities and humanitarian organisations to anticipate potential crises and allocate resources efficiently. This comprehensive approach not only strengthens the ability to manage emergencies more effectively but also contributes to identifying underlying links and patterns between climate-induced displacement and disease spread, informing long-term preventive strategies. Two projects starting in Burkina Faso and Ethiopia [29,30] are examples of these integrated approaches, where they plan to implement surveillance methods in line with the International Health Regulations.

*Coordinated International Collaboration*: Promotes a more integrated international coordination, where international public health and disaster management agencies work hand in hand to share information, resources, and experiences in real time. We are aware that cyclones, hurricanes, and typhoons simultaneously impact multiple countries in Central America and Southeast Asia each season. We also know that this will continue to occur every year. In an increasingly interconnected world, fostering collaborative efforts on an international (regional) scale is essential to address complex challenges effectively. By encouraging collaboration between global health and disaster management organisations, we can establish a more comprehensive approach to crisis response. This involves the sharing of critical information about disease outbreaks and climatic events, pooling resources to ensure a more efficient and targeted response, and exchanging best practices and lessons learned. Integrated international coordination not only enhances the ability to respond

promptly to emergencies but also builds a collective reservoir of knowledge and expertise. This collaborative synergy allows for a more cohesive global response, minimising the duplication of efforts and maximising the impact of interventions. By breaking down silos and fostering a culture of shared responsibility, we can collectively navigate the intricate landscape of health and disaster management, enhancing our collective resilience to unforeseen challenges. This approach ensures that countries and communities are better equipped to face the dual challenges of climate-induced displacement and disease outbreaks in a more synchronised and effective manner. In this regard, the Sendai Framework for Disaster Risk Reduction 2015–2030 is the key international framework to drive international cooperation, as reported by the mid-term evaluation conducted by Australia [31].

Anticipatory humanitarian funding options and strengthening protection and human rights: A component of debt relief is crucial, particularly as the costs of mitigating disaster displacement during the pandemic prove inevitably higher [32]. Simultaneously, reinforcing the protection and human rights of people on the move is essential. Governments must uphold obligations under international law, safeguarding the rights and welfare of internally displaced people and internal migrants [33–36]. The goal is to establish a form of legal protection similar to that for refugees but applicable to individuals moving within their own country [37].

*Adaptive Quarantine Protocols*: Establishing quarantine and isolation protocols that are adaptable to displacement contexts, recognising the mobility of affected populations and ensuring continuous medical care. The proposal for Adaptive Quarantine Protocols aims to address the complexity of displacement situations by establishing flexible guidelines that cater to the specific needs of affected populations. Recognising the inherent mobility of displacement, these protocols should be designed to be applicable in dynamic contexts, where communities may be constantly on the move due to extreme climate events or other emergencies. Adaptability involves considering the availability of resources, healthcare infrastructure, and the continuity of medical care, as implemented by Ukraine in 2021 [38]. Additionally, sociocultural characteristics of displaced populations must be considered to ensure that protocols are culturally sensitive. This measure not only aims to manage the spread of diseases but also to ensure the overall well-being of displaced communities by providing continuous medical care and addressing specific needs that arise in displacement contexts.

*Integrated drills*: Conduct integrated drills that replicate realistic scenarios where displacement and pandemics coexist. This involves creating simulated situations that mirror the challenges posed by both displacement events and pandemics simultaneously. The purpose of these integrated drills is to assess the effectiveness of integrated responses and make necessary adjustments to plans. Furthermore, these simulations can be implemented by using artificial intelligence tools [39]. By incorporating elements of both displacement and pandemic response into these simulations, emergency responders and relevant authorities can assess the coordination, communication, and overall efficacy of their strategies. This approach not only helps identify potential gaps or weaknesses in the existing response frameworks but also facilitates the refinement of protocols to enhance overall preparedness. Furthermore, integrated drills provide an invaluable opportunity for different agencies, including health departments, disaster management teams, and humanitarian organisations, to collaborate and streamline their efforts. This collaborative approach can foster a better understanding of how various sectors can work together seamlessly during complex emergencies, improving the overall resilience of communities facing the dual challenges of displacement and pandemics. The insights gained from these simulations contribute to continuous learning, ensuring that response strategies remain adaptive and robust in the face of evolving threats and crises.

## 4. Discussion

The global impact of 50.2 million internal displacements during 2020–2021 due to severe climate events [4] highlights the pressing need for a legal framework. International

agreements such as the Paris Agreement, Sendai Framework, and Global Compact for Migration provide a foundation for coordinated global responses [10,11], yet adaptation to health crises is a challenge.

The existing legal frameworks, including the Guiding Principles on Internal Displacement [7], must be tailored to health crisis contexts. The emphasis on intensifying actions, promoting early intervention, and protection against discrimination aligns with the evolving nature of displacement crises.

Legal preparedness should facilitate intersectoral coordination, addressing gaps between health and disaster management sectors. Ensuring financial support for local adaptation and integrating climate finance within legal frameworks is imperative for effective resource utilisation and synergies between humanitarian, development, and climate funds.

In response to these challenges, the proposed response policies provide a roadmap for mitigating the impact of dual challenges. Integrated response centres, contingency plans, and surveillance systems address immediate needs, emphasising the importance of collaborative efforts. Anticipatory humanitarian funding, adaptive quarantine protocols, and integrated drills contribute to long-term resilience.

The nexus of climate-induced displacement, pandemics, and legal preparedness demands a nuanced approach. Strengthening legal frameworks, fostering international collaboration, and integrating health and displacement responses are critical. The identified vulnerabilities underscore the urgency of tailored policies, ensuring a holistic and adaptive strategy to safeguard displaced populations amidst evolving challenges.

The comprehensive analysis of the intersection between internal displacement, extreme climate events, and pandemics underscores the complexity of managing emergencies impacting displaced populations. The assessment of existing legal frameworks reveals that, while there are robust international and regional instruments for the protection of internally displaced persons, there are significant gaps in their applicability during health crises.

Policy recommendations emphasise the need to adapt and strengthen legal frameworks, considering specific protocols for displacement scenarios during pandemics. The importance of integrated international coordination, where public health and disaster management sectors collaborate effectively, is highlighted. Practical strategies, such as the implementation of integrated drills and the establishment of adaptable quarantine protocols, are proposed.

The connection between climate-related displacement and disease spread, as evidenced by the COVID-19 pandemic, underscores the urgency of integrated approaches. The adaptation of the Guiding Principles on Internal Displacement and the consideration of region-specific legal frameworks are essential. The implementation of integrated surveillance systems provides an effective tool for monitoring and responding to the dynamics of these interconnected phenomena.

Among the limitations of this study, we must mention the lack of variables related to the duration of displacement, which hinders analysis of displacement periods and potential exposure to COVID-19 contagions during emergencies [22,24]. Another limitation is related to the lack of other sources of data for producing accurate estimations on internal displacement or for triangulation purposes with existing data sources. We are aware of the limitations of a single discourse, considering the multiplicity of interests involved in the management of extreme weather events and pandemic outbreaks, which makes it difficult to structure a coordinated global agenda among the different actors from both the public and private sectors. In this study, we were not able to analyse the effects of policy stringency on COVID-19 spread and the impact of these policies on internally displaced populations. On the one hand, the impacts of the policies have been so varied that no single article comprehensively analyses all findings. There are separate epidemiological, economic, and mental health studies, for example. Notably, the impact of policies varied significantly depending on the cultural context of each country or subnational unit, as illustrated by a previous study [40].

## 5. Conclusions

We can conclude that addressing the complexity of the intersection between displacements, extreme climate events, and pandemics requires a global, coordinated, and adaptable response. Preparedness, adaptation of legal frameworks, and collaboration between sectors and countries are crucial for effectively managing emergencies and ensuring the protection and care of internally displaced persons in health crises.

The analyses and proposals presented in this work serve not only as an endpoint but as a precursor for further, more targeted inquiries. The identified complexities, gaps, and recommendations function as guiding signposts, directing attention toward the necessity of ongoing research endeavours. The call for adaptability, collaboration, and preparedness necessitates a continual exploration of these themes to foster a deeper comprehension of the global, coordinated, and adaptable response required in navigating the intricate landscape of health crises.

**Author Contributions:** R.A.A.Z., G.N.d.L. and J.C.S.-H. contributed to the conception of the work, the analysis, and the interpretation of data. All authors have drafted the work and approved the submitted version. R.A.A.Z., G.N.d.L. and J.C.S.-H. agree to be personally accountable for the author's contributions and for ensuring that questions related to the accuracy or integrity of any part of the work, even ones in which the author was not personally involved, are appropriately investigated, resolved, and documented in the literature. All authors have read and agreed to the published version of the manuscript.

**Funding:** This research received no external funding. The APC was funded by the University of Helsinki Library. Open access funding provided by University of Helsinki.

**Data Availability Statement:** Publicly available datasets were analysed in this study. This data can be found here: https://gidd.idmcdb.org/ (accessed on 1 December 2023).

**Conflicts of Interest:** The authors declare no conflicts of interest.

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
