# Peer review of "Tackling Complexity: Integrating Responses to Internal Displacements, Extreme Climate Events, and Pandemics"

_climate, doi:10.3390/cli12030031_

Round 1

Reviewer 1 Report

Comments and Suggestions for Authors

This study estimated the internal displacements during 2020 and 2021 due to climate-related events, and to review the evidence for proposing policy recommendations. During 2020 and 2021, over 50.2 million people worldwide were forced to leave their homes due to climate related disasters, mainly concentrated in developing countries, exacerbating health risks. The article analyzes existing legal frameworks, such as the Guiding Principles on Internal Displacement, emphasizing the need to adapt to health crisis situations. At the same time, policy recommendations were put forward, including establishing a comprehensive response center, developing integrated emergency plans, implementing integrated monitoring systems, and strengthening international coordination. This work is very meaningful and most sustainable development scholars are also interested, However, there are the following questions that are not clear to me, Climate is a scientific, peer-reviewed, open access journal of climate science, I have always believed that the scope of this journal is more focused on the phenomena, patterns, processes, and methods of climate itself, while the focus of this article is not on climate. Climate seems to be just an introduction. If the scope of publication is appropriate, then, I only have one question: how did the author conclude that displaced people are caused by climate factors, and I did not see a detailed method statement in the article.

Comments on the Quality of English Language

No.

Author Response

Dear Reviewer: Thank you very much for your insightful comments on the manuscript. We have thoroughly reviewed the journal's scope and found that it states: "Impact of climate on economy and society". Therefore, we believe that our manuscript is a good fit for the journal's scope.

Regarding the language editing, please, note that we are using British English style in our manuscript. It was completely proofread by a native English Speaker.

Reviewer 2 Report

Comments and Suggestions for Authors

Review report attached 

Author Response

Review

  • The paper is clear in the methodology, it is relevant because it aimed a period of time under covid-19 lockdowns (2020-2021), considering the relationship between climate, disasters, relocation and a covid scenario.

Dear Reviewer,

We would like to express our sincere gratitude for the time and effort you dedicated to reviewing our manuscript. Your insightful comments proved invaluable in helping us improve its quality.

  • The 37 cited references are recent publications, it does not include an excessive number of self-citations.

R: Thank you for your comments.

  • I consider it is necessary to include some privately funded research that does not come from some type of government funding databases to prove a true dichotomy in the use of the data. In the methodology, references are made to government databases which is very significant, since governments have opinions, and power agendas. I suggest it be revised, especially to avoid repeating events that led to greater control over individuals and populations.

Given this, I recommend a balanced view, about the facts. I recommend including some recognized private studies of scientists in the field like Patrick J. Michaels. He was a contributing author and was a reviewer of the United Nations Intergovernmental Panel on Climate Change, which was awarded the Nobel Peace Prize in 2007, and his last book “Scientocracy” reveals the urgency of including private data to balance the government data.

R: Actually, the dataset used is from the Internal Displacement Monitoring Centre (IDMC), which is affiliated with the Norwegian Refugee Council, a non-governmental organisation (NGO). We acknowledge certain limitations present in the data, which have been previously analysed by the authors, and we address these limitations in the Discussion section. However, to the best of our knowledge, no private company possesses the IDMC's capacity for gathering global data on internal displacement. We are currently developing alternative data collection techniques, such as web scraping methods to gather "in vivo" data from news articles. However, this is an ongoing and separate study conducted by our research team, to be published in due course.

To address your concerns, we have included the following paragraphs:

In the Methods section:

The IDMC is the leading NGO for collecting, processing, curating, and publishing information on internal displacement due to disasters and conflicts globally. Their methods and terminology have been clearly described in their website [4]. The IDMC prioritizes its data sources, with the first level comprising data primarily produced by government agencies, UN organizations, the International Organization for Migration (IOM), humanitarian clusters, and local authorities. The second level includes data from international and local NGOs, civil society, human rights organizations, and academia. Finally, the third level contains data from international and local media, affected pop-ulations, and non-state armed groups [4]. The first level sources are considered the most reliable and are used as the primary basis for producing estimates. The second level sources are consulted only when information from first-level sources is unavailable. The third level sources are only used if their reliability can be rigorously assessed and verified [4].

In the Discussion section:

Among the limitations of this study, we must mention the lack of variables related to the duration of displacement, which hinders analysis of displacement periods and potential exposure to COVID-19 contagions during the emergencies [20,22]. Other limitation is related to the lack of other sources of data for producing accurate estimations on internal displacements, or for triangulation purposes with existing data sources. We are aware of the limitations of a single discourse considering the multiplicity of interests involved in the management of extreme weather events and pandemic outbreaks, which makes it difficult to structure a coordinated global agenda among the different actors from both the public and private sectors. In this study we were not able to analyse the effects of policy stringency on COVID-19 spread and the impact of this policies in internally displaced populations. On the one hand, the impacts of the policies have been so varied that no single article comprehensively analyses all findings. There are separate epidemiological, economic, and mental health studies, for example. Notably, the impact of policies varied significantly depending on the cultural context of each country or subnational unit, as illustrated by a previous study [38].

  • The experimental design is appropriate to test the hypothesis from Websites of governmental agencies and international organizations (WHO/WHO, PAHO/PAHO, UNISDR/ISDR), for a governmental approach to the internal displacements and disaster situations. The results are reproducible according to the details given in the methods section, taking the same sources: Papers, book chapters, institutional reports during 2020-2021.

R: Thank you for your comment.

  • Map 1 is appropriate and shows the data in an easy form to interpret and understand, showing that China, India and the US have the largest internal displacement resulting from climate events. Table 1 is also easy to understand and identify the main 20 climate - related events.

R: Thank you for your comment.

  • The government databases evidence presented shows 50.2 million internal displacements during 2020-2021 due to severe climate events, but the need for a legal framework as conclusions requires suggestions based on previous experiences, the covid Lockdown and the suggested WHO policies were counterproductive, they affected individuals and their freedoms more, since they focused on strengthening the interests of governments, without managing to effectively reduce the number of global deaths.

As mentioned in our previous comments, an NGO maintains and curates the dataset.

Regarding the effects of policy stringency on COVID-19 spread, the results have been so varied that no single article comprehensively analyses all findings. There are separate epidemiological, economic, and mental health studies, for example. Notably, the impact of policies varied significantly depending on the cultural context of each country or subnational unit, as illustrated by this study: “Quote".

  • The review es clear , it is relevant for the climate change discourse, it finds a gap in knowledge between the climate events and the internal displacements during covid timeline, but It avoids some of the private data that focus on questionable facts about the Climate rhetoric, assuming the UN discourse as a neutral and a valid source, not showing the complexity of interest behind each of the legal laws, many of them which may affect as covid lockdowns more than provide a coordinated global response.

We are aware of the limitations of a single discourse considering the multiplicity of interests involved in the management of extreme weather events and pandemic outbreaks, which makes it difficult to structure a coordinated global agenda among the different actors from both the public and private sectors. Therefore, in the subsection Intersectoral Coordination Mechanisms we discuss the importance of a coordinated approach across sectors, such as public health and disaster management, which may be absent in the current legal framework.

  • If there exists any political intention of acting above individual response and liberty, the legal framework may not be the best way to respond before these challenges, it would be better to let local people decide the proper way of action. The climate science is relatively new, just 100 years ago was created, but as weapon discourse can be used to propose lasting changes in international law and institutions, so I recommend do not use it to be politicized in a global level, the climate science does not have to become a political issue, or a surveillance system based on the promises of provide a roadmap for mitigating the impact of future challenges.

We promote a balance between, on the one hand, a legal framework based on the principles of participatory inclusive governance, and on the other hand, strategic alliances that take shape in the territories between the social actors involved in the structuring of a coordinated global agenda (Subsection Adaptation to Local and Mobile Contexts, “These regulations recognize the need to adapt to specific contexts and the mobility of populations”).

Additionally, we state: Integrated international coordination not only enhances the ability to respond promptly to emergencies but also builds a collective reservoir of knowledge and expertise.

  • The paper is well defined, but if it is possible, I suggest focusing on some concepts related to the current knowledge, e.g., Weather: is what you see every day and climate is what you expect to see every day. The weather changes day to day, but climate can change over more periods of time. Environment is everything that surrounds the living; ecology is the relationship of living things with each other in their environment.
  1. Thank you for your observation.

Taking into consideration the concepts most relevant for a comprehensive understanding of the analysis, we have incorporated the following paragraphs at the beginning of the Results Section:

"The analysis of data from the Internal Displacement Monitoring Centre and the bibliographic review addressing responses to internal displacements in climate-related disasters has yielded relevant results. However, it is important, initially, to define some of the concepts discussed for a comprehensive understanding of the results.

Climate change refers to long-term alterations in global climate patterns, primarily resulting from human activities releasing greenhouse gases into the atmosphere. These changes have significant impacts on climate, such as rising average temperatures, alterations in rainfall patterns, and more frequent extreme weather events, directly influencing ecosystems, biodiversity, and the daily lives of communities.

Extreme climate events, in turn, are intensified and uncommon episodes within the climate system. They encompass phenomena like severe storms, heatwaves, floods, wildfires, and hurricanes, causing significant impacts on the environment, infrastructure, and society. These events often lead to natural disasters, population displacements, loss of human lives, and substantial damage to ecosystems. Understanding these concepts is crucial for addressing the complex and interconnected challenges related to the environment and climate, seeking adaptation and mitigation strategies to ensure the resilience of communities in the face of these adverse scenarios."

We believe that these additional paragraphs provide a more comprehensive foundation for readers to grasp the key concepts underlying our analysis. We also checked these terms throughout the manuscript.

  • The results are significant, they show complexity in the results, pointing to support the proposal of a global coordinated and adaptive response, based on adaptation of legal frameworks, under the rhetoric of protection and care.

R: Thank you for your comment.

The article is written in an appropriate way, with proper analyzes of climate disasters and relocation. The paper will attract a wide readership because it integrates covid scenario, displacement of people, metric of disasters and legal framework for global interventions. The level of English is appropriate and understandable.

R: Thank you for your comment.
